# A Spotlight on the Egyptian Honeybee (*Apis mellifera lamarckii*)

**DOI:** 10.3390/ani12202749

**Published:** 2022-10-13

**Authors:** Hesham R. El-Seedi, Aida A. Abd El-Wahed, Chao Zhao, Aamer Saeed, Xiaobo Zou, Zhiming Guo, Ahmed G. Hegazi, Awad A. Shehata, Haged H. R. El-Seedi, Ahmed F. Algethami, Yahya Al Naggar, Neveen F. Agamy, Mostafa E. Rateb, Mohamed F. A. Ramadan, Shaden A. M. Khalifa, Kai Wang

**Affiliations:** 1International Research Center for Food Nutrition and Safety, Jiangsu University, Zhenjiang 212013, China; 2Pharmacognosy Group, Department of Pharmaceutical Biosciences, Biomedical Centre, Uppsala University, P.O. Box 591, SE-751 24 Uppsala, Sweden; 3Department of Chemistry, Faculty of Science, Menoufia University, Shebin El-Koom 32512, Egypt; 4International Joint Research Laboratory of Intelligent Agriculture and Agri-Products Processing, Jiangsu Education Department, Nanjing 210024, China; 5Department of Bee Research, Plant Protection Research Institute, Agricultural Research Centre, Giza 12627, Egypt; 6College of Marine Sciences, Fujian Agriculture and Forestry University, Fuzhou 350002, China; 7Department of Chemistry, Quaid-i-Azam University, Islamabad 45320, Pakistan; 8School of Food and Biological Engineering, Jiangsu University, Zhenjiang 212013, China; 9Zoonotic Diseases Department, National Research Centre, Giza 12622, Egypt; 10Avian and Rabbit Diseases Department, Faculty of Veterinary Medicine, University of Sadat City, Menoufia 22857, Egypt; 11PerNaturam GmbH, An der Trift 8, 56290 Gödenroth, Germany; 12Prophy-Institute for Applied Prophylaxis, 59159 Bönen, Germany; 13Faculty of Medicine, Riga Stradins University (RSU), LV-1007 Riga, Latvia; 14Alnahal Aljwal Foundation Saudi Arabia, P.O. Box 617, Makkah 24211, Saudi Arabia; 15Zoology Department, Faculty of Science, Tanta University, Tanta 31527, Egypt; 16Nutrition Department, Food Analysis Division, High Institute of Public Health, Alexandria University, Alexandria 21561, Egypt; 17School of Computing, Engineering and Physical Sciences, University of the West of Scotland, Paisley PA1 2BE, UK; 18Central Agriculture Pesticides Laboratory, Pesticide Analysis Research Department, Agriculture Research Center, Giza 24221, Egypt; 19Department of Molecular Biosciences, The Wenner-Gren Institute, Stockholm University, SE-106 91 Stockholm, Sweden; 20Institute of Apicultural Research, Chinese Academy of Agricultural Sciences, Beijing 100093, China

**Keywords:** beekeeping, beehives, Egyptian honeybee (*Apis mellifera lamarckii*), genetic analysis, defensive behaviors

## Abstract

**Simple Summary:**

The Egyptian honeybee (*Apis mellifera lamarckii*) is one of the honeybee subspecies known for centuries since the ancient Egypt civilization. The subspecies of the Egyptian honeybee is distinguished by certain traits of appearance and behavior that were well-adapted to the environment and unique in a way that it is resistant to bee diseases, such as the Varroa disease. The subspecies is different than those found in Europe and is native to southern Egypt. Therefore, a special care should be paid to the vulnerable *A. m. lamarckii* subspecies and greater knowledge about the risk factors as well as conservation techniques will protect these bees. Additionally, more qualitative and quantitative measures will be taken to obtain deep insights into the *A. m. lamarckii* products’ chemical profile and biological characters.

**Abstract:**

Egypt has an ongoing long history with beekeeping, which started with the ancient Egyptians making various reliefs and inscriptions of beekeeping on their tombs and temples. The Egyptian honeybee (*Apis mellifera lamarckii*) is an authentic Egyptian honeybee subspecies utilized in apiculture. *A. m. lamarckii* is a distinct honeybee subspecies that has a particular body color, size, and high levels of hygienic behavior. Additionally, it has distinctive characteristics; including the presence of the half-queens, an excessive number of swarm cells, high adaptability to climatic conditions, good resistance to specific bee diseases, including the *Varro disorder*, and continuous breeding during the whole year despite low productivity, using very little propolis, and tending to abscond readily. This review discusses the history of beekeeping in Egypt and its current situation in addition to its morphology, genetic analysis, and distinctive characters, and the defensive behaviors of native *A. m. lamarckii* subspecies.

## 1. Introduction

Beekeeping has been conducted for thousands of years; hence, the honeybee and their products have been known in ancient Egypt. Egyptians used bee products in daily life, including honey and wax, as food and in ceremonies. They also recorded their knowledge and practices on templates and tombs [1,2].

The honeybee subspecies *Apis mellifera lamarckii* is assumed to be the same species as the one discovered during the Pharaohs’ period [3,4]. *A. m. lamarckii* Cockerell, 1906 is geographically distributed in the Egyptian desert, delta region, Nile valley, and Sudan [5,6,7]. Lamarck’s honeybee was previously known as *Apis fasciata* Latreille and is considered an offshoot of adansonii. It resembles the lighter, yellow-banded varieties with very dark drones, and the width of the worker cells is identical to that of adansonii from center to center [7]. Lamarck’s bee is described as a good housekeeper but a poor honey producer. Thus, the Carniolan honeybee, which gained popularity because it is peaceful and easy to manage in modern Langstroth hives, thus effectively replaced the *A. m. lamarckii* in commercial beekeeping in Egypt. 

As a result, the native honeybee population in Egypt was suppressed and predominantly centered in the Manfalut area of the Assiut governorate, which was surrounded by hybrids from Europe. *A. m. lamarckii* was introduced to the Dakhla oasis in 1928, and was utilized there until 1960, when the New Valley Government decided to breed less aggressive bees than *A. m. lamarckii* [8,9]. A large number of *A. m. lamarckii* colonies (about 400,000) were found in traditional mud-tube hives in the Assiut Governorate and small populations in isolated oases of Egypt [10]. The *A. m. Lamarckii* is noticeably smaller and has legs and wings that are shorter. Compared to the European subspecies, its colonies have fewer bees. It does not store food for the winter or form winter clusters, and it does not stop reproducing for practically the entire year. It is viewed as a representative model of the tropical African bees in general [11].

The phylogenetic analyses of *A. m. lamarckii* and other subspecies demonstrated a close relationship between *A. m. lamarckii* and *A. m. syriaca* (Syrian honeybee) [12]. *A. m. lamarckii* has various distinctive characteristics, including a higher quality of reared queens [10], *A. m. lamarckii* exerts high levels of defensive behavior [13]. The Egyptian honeybee exhibits a more combative defensive response when compared to European bees, and the *A. m. lamarckii* larva develops more quickly; hence, they tolerate *Varroa destructor mites*, the serious and dangerous pest to *Apis mellifera* species [9]. Additionally, the biochemical and molecular characterization for three subspecies of honey bee workers explained the high genetic gap of 0.25 separating the Egyptian *A. m. lamarckii* subspecies from the other two subspecies, *A. m*. *ligustica* (Italian), and *A. m. carnica* (Carniolan) [14]. Here, we aimed to shed the light on the history of beekeeping in Egypt and the Egyptian honeybee subspecies *A. m. lamarckii*, reviewing their morphology, genetic analysis, distinctive characters, and defensive behaviors as part of our efforts to study honeybee and its bee products [15,16,17,18,19,20,21,22,23,24].

## 2. The History of Beekeeping in Egypt and Its Current Situation

Honey is a natural product produced by bees and used in Egypt not only as a sweetener but also associated with medical practices, and was deemed a poison for ghosts, demons, evil spirits, and the dead, representing a symbol of resurrection [25]. The first crude examples of the honeybee hieroglyphs were carved by the Egyptians of the first dynasty, in approximately 3000 B.C.E [2,3,4,26]. In the Nile Valley region, the bees were used as a source of honey from the earliest years; thus, bees were highly appreciated insects by the ancient Egyptians. In the old kingdom, the earliest inscription exemplifying beekeeping came from the sun temple of pharaoh Newossere in the fifth dynasty and back to 2450 B.C.E. [27]. In the sun temple, a room adjacent to the central obelisk was discovered by Ludwig Borchardt in 1898 and called “The Chamber of the Seasons” as it contains reliefs of activities that happened at particular times of the year, and one of them was found to be the oldest evidence of beekeeping [1,27]. The bas-relief from left to right shows four scenes: (I) a beekeeper working with the beehives; (II) three men pouring honey into containers; (III) two men processing honey (this scene is mostly missing); (IV) a beekeeper sealing honey in a vessel for storage [1]. By the end of the old kingdom and during the sixth dynasty, honey production increased to the level of trading [26]. During the time of the new kingdom, there were many tombs with images illustrating the practices involved in the treatment using bees and their products [28]. In the tomb of the 18th Dynasty vizier Rekhmire, there were inscriptions demonstrating honeycombs gathering from large horizontal hives, as shown in Figure 1, pouring the honey into large vessels, the successive honey sealing in diamond-shaped containers, and comb pulverizing [1]. During the 26th dynasty, the tomb of Pabasa demonstrated one of the most famous beekeeping reliefs in Egypt, where a beekeeper is facing a group of honeybee and a series of horizontal hives with his hands held up in praise. These horizontal hives were similar to the carved hives of the old kingdom from Newossere Any’s sun temple, as shown in Figure 2. They also authenticated the continued value of the honey and the honeybee in the ancient Egyptian times and the progress of hives types throughout the time [1,29,30].

In the Ptolemaic Period (304–30 B.C.E.), the state taxed bee keeping and the bee derived products. In the Nile Valley, beekeeping processes and breeding programs were established, as honey was fundamental to the people’s food [25]. The Egyptian mud hives (traditional bee hives) were placed in piles that could reach hundreds and were combined by pouring mortar in-between [31,32]. In 1918, modern beekeeping started in Egypt using wooden Langstroth frames. In 1920, the first association of beekeepers was established to improve beekeeping process and develop its marketing. Since, then traditional beekeeping methods were used in parallel with modern ones.

The modern beekeeping represents 99% of the used practices in Egypt [9]. Figure 3 illustrates the difference between the Egyptian mud traditional hives (A1-3) and the modern wooden ones (B). According to the Food and Agriculture Organization (FAO), statistics on the number of beehives in Egypt between 1961–2020 have been fluctuating, as represented in Figure 4. The number of beehives demonstrated noticeable increases between 1964–1972, 1980–1990, and 1999–2001, reaching 937,000, 1,651,000, and 1,485,000 hives, respectively, at the end of each period. In contrast, they demonstrated noticeable decreases in 1980, 1994, and 2017, reaching 858,000, 1,225,000, and 820,516 hives, respectively [9].

Honey bees are given special attention in Egypt because of their importance in pollination and their impact on the economy [33]. The pollination is mainly conducted using the Egyptian clover blooming during June, cotton flowering during August–September, and a minor contribution of citrus in April [31,32]. In the future, thermal stress on the Egyptian honey bee colonies will be a significant problem for beekeepers, especially during summer [24,34].

## 3. *Apis mellifera lamarckii* Morphology

The Egyptian *A. m. lamarckii* is considered an offshoot of *adansonii*. [7]. Some morphometric characteristics of Egyptian *A. m. lamarckii*, *African Apis mellifera scutellata,* and *Apis mellifera jemenitica* were reported where these subspecies were very close to *A. m. lamarckii,* as mentioned in Table 1, using different techniques such as an electron microscope [7,11,35,36,37,38,39]. Otherwise, the average length of the honeybee forewing in difference races was reported. The highest average was 10.700 mm in *A. m. florea*, while the lowest average was 8.275 mm in *A. m. lamarckii*. The average head length on different races in adult gave the highest average of 5.575 mm in *A. m. lamarckii*, while the lowest average was 3.750 mm in *A. mellifera ligustica* [38]. The mean body mass of the European (*Apis mellifera Carnica*) bee amounted to 120 mg, while the Egyptian bee to 78 mg. This shows that the Egyptian bee is much smaller than the European one, with only 65% of its mass. Similar results related to the thorax masses of both subspecies differ significantly (33 mg vs. 24 mg) for the European and Egyptian bee, respectively. The Egyptian bee is characterized with shorter wing size (45 mm) compared to the European bee (55 mm). The Egyptian honeybee *A. m. lamarckii* is significantly smaller, slimmer, and has shorter wings and legs. Moreover, the wing load was 17 and 21 Nm^−2^ for the Egyptian and European bee, respectively [11].

The Egyptian honeybee *A. m. lamarckii* neither forms winter clusters nor stores food for overwintering and breed nearly throughout the year. *A. m. lamarckii* is forced to form winter clusters and to store food for unfavorable periods [11].

## 4. Distinctive Characters of *Apis mellifera lamarckii*

The Egyptian *A. m. lamarckii* is known for its excessive large number of swarm cells, continuous breeding during throughout the whole year, low productivity using very little propolis, and proclivity to migrate, tending to migrate readily [7,11]. Its colonies include fewer bees (8000–10,000) compared to the European subspecies, which have more than 40,000 bees. *A. m. lamarckii* was found to exhibit a higher mass specific metabolism than the European *A. m. carnica*, and *A. m. lamarckii,* being more aggressive, active, and well adapted to its environment [11,47]. Another characteristic of *A. m. lamarckii* is the presence of half-queens, which are intermediate in morphology between queens and workers. They have a number of ovarioles that range between workers and queens, are possibly egg-laying workers, and help the queen’s colony by contributing to egg laying. *A. m. lamarckii* flagella of workers were longer than those of queens and half-queens; however, they did not differ in the length of flagellum. [38,48]. A study has been conducted to evaluate the quality of reared queens produced from *A. m. carnica* (Carniolian honey bee), *A. m. ligustica* (Italian honey bee), and *A. m. lamarckii* colonies. The highest accepted percentage of *A. m. lamarckii*-grafted larvae was significantly higher than *A. m. ligustica*, where the acceptance % was (93.75, 87.2, and 62.5) in *A. m. carnica*, *A. m. lamarckii* and *A. m. ligustica*, respectively. The Egyptian and the Carniolian colonies have a higher quality of reared queens represented in the long queen cell size, heaviest virgin queen, large number of ovarioles, and large volume of spermatheca of the produced queens than that of the Italian honey bees [10]. On the other hand, *A. m. lamarckii* was more resistance to chalkbrood diseases, with an infestation average of 0.218 % after the three inoculations. In contrast, the Carniolan race (*A. m. carnica*) demonstrated a lower tolerance, with an infestation average of 0.844% [49].

## 5. Genetic Analysis of *Apis mellifera lamarckii*

Using various analytical tools and different approaches, *A. m. lamarckii* mtDNA sequences were analyzed for a better understanding of its phylogenetic relationships with other different subspecies as well as to detect the genetic characteristics of these bees. The phylogenetic analyses of *A. m. lamarckii* and other different subspecies demonstrated a close relationship between *A. m. lamarckii* and *A. m. capensis*, *A. m. intermissa*, *A. m. ligustica*, and *A. m. scutellata*. The highest identity percentage (95.78%) was between *A. m. lamarckii* and *A. m. syriaca,* whereas *A. m. lamarckii* were far from *A. florea*, *A. koschevnikovi*, *A. cerana japonica*, *A. laboriosa*, *A. nuluensis*, and *A. m. sahariensis* [12]. Another study confirmed the close phylogenetic relationship between *A. m. lamarckii* and *A. m. syriaca* using the complete mtDNA sequences [50]. The honeybee in the dry regions of Sudan (desert, semi desert, and dry savannah) were genetically similar to *A. m. lamarckii* and *A. m. syriaca,* as both of them possess the O-lineage [51]. Abou-Shaara et. al. [52] confirmed a close genetic relationship between *A. m. lamarckii*, *A. m. jemenitica* (Arabian or Nubian honeybee), and *A. m. syriaca*. *A. m. lamarckii* was distinguished by the restriction of the Hinf-I enzyme from other honeybee subspecies using a PCR assay [53]. CO I/Hinf I was used not only to discriminate the mitochondrial “O” lineage but also as a diagnostic site for *A. m. lamarckii* [54]. The whole mitochondrial genome of the Egyptian *A. m. lamarckii* was examined and a 16,589 bp mitochondrial genome of *A. m. lamarckii* including 37 classical eukaryotic mitochondrial genes and an A + T-rich region were revealed. The arrangements and directions of the genes were identical to those of other *Apis* mitogenomes. All genes terminated with TAA, seven genes started with ATT, four with ATG, and two with ATA. Nine genes were encoded on the light strand and four on the heavy strand. The whole 22 tRNA genes had a cloverleaf structure that ranged from 66 to 80 bp. *A. m. lamarckii* is clustered with other *A. mellifera* subspecies in the phylogenetic tree [55]. A study has been conducted on two *Apis mellifera* (*A. m. lamarckii* and *A. m. carnica*) races as well as the hybrid and demonstrated the presence of clear genetic variations between *A. m. lamarckii* and *A. m. carnica*; the hybrid was closely related to the Carniolan (*A. m. carnica*) rather than to the *A. m. lamarckii* race. The higher resemblance between the Carniolan race and the hybrid suggested that the hybrid originated from Egyptian drones and Carniolan queens [56]. The genetic relationships and identical genetic characteristics between different African bee subspecies were investigated and demonstrated that *A. m. lamarckii* has the least genetic relationships with all other studied African bee subspecies (*A. m. scutellata*, *A. m. capensis*, *A. m. intermissa*, and *A. m. monticola*). While those of north and east Africa, namely, *A. m. intermissa* and *A. m. monticola*, exhibited less genetic similarity, the African subspecies *A. m. scutellata* demonstrated considerable genetic similarity and a tight evolutionary link to *A. m. capensis*. There were very few similarities and no close links among the bees from North Africa, *A. m. Intermissa*, and *A. m. Lamarckii* [57].

## 6. *Apis mellifera*
*lamarckii* Hygienic Behaviors

The hygienic behavior of different honeybee species plays a critical role in the colony’s health of diverse honeybee races [58,59]. The hygienic behavior represents a behavioral defensive response of honeybee workers to uncap the wax covering of the brood cells, detect diseased brood, get rid of infected larvae or pupae, and inhibit the spreading of infective diseases [60,61]. Hygienic behaviors enable bees to resist parasitic mites (*Varroa destructor*) [60,62], American foulbrood (*Paenibacillus larvae*), and chalkbrood (*Ascosphaera apis*) among other infestations [63,64]. Hygienic behavior helps maintain the health of densely populated insect societies by limiting the horizontal transmission of pathogens and population growth of parasites [65]. Hygienic behavior relies on a limited set of genes linked to different regulation patterns associated with an over-expression of cytochrome P450 genes [66].

Native honeybee races (for example, *A. m. jemenitica* and *A. m. syriaca*) exhibited a certain degree of hygienic behavior compared to the exotic one, owing to its different genetic structure as well as its compatibility with local environmental conditions [67,68]. The *A. m. jemenitica* native race was significantly higher than that of the exotic one, *A. m. carnica.* The deformed mites, as a result of the grooming behavior, were also significantly higher in the colonies of the native race than those of exotic one [68]. Allam and Zakaria reported two defensive behavior mechanisms against the Varroa mite, namely the hygienic and grooming behaviors to minimize the parasitic threat. Moreover, the hygienic behavior of the Egyptian hybrid honeybee represented in cutting up Varroa mite bodies was related to treating the diseased colonies with black cumin oil or an oils mixture [69,70]. *A. m. lamarckii* are characterized by their high levels of defensive behaviors [13]. *A. m. lamarckii* colonies were found to exhibit higher levels of hygienic behavior compared to the Egyptian *A. m. carnica* colonies, according to a study conducted by Kamel and other collaborators [13]. The study demonstrated that, after 24 h, the dead broods were totally removed from 42.9% of *A. m. lamarckii* colonies, while 0% of the Egyptian *A. m. carnica* colonies were free from the dead broods. After 48 h, 71.4% of *A. m. lamarckii* colonies were completely free from the killed broods, whereas 8.3% of the *Carniolan hybrid A. m. carnica* colonies cleaned all of the killed broods [13].

In a different study, the percentages of cleaned cells were determined at different times for the local *A. m. lamarckii* and the Carniolan hybrid colonies and demonstrated 32.2% and 15.5% after 6 h, 70.6% and 38.4% after 24 h, 83.8% and 46.7% after 30 h, and 93.4% and 64.4% after 48 h, for the local *A. m. lamarckii* and the Carniolan hybrid colonies, respectively [35]. Clearly, the results revealed that *A. m. lamarckii* has significantly higher levels of hygienic behaviors, which could be affecting the survival and efficiency of the whole honeybee colonies against the Varroa mite infestation [13,35,60,65].

## 7. Biological Properties of Bee Venom from *A. m. lamarckii*

The venom produced by the Egyptian honeybee has a biological activity and potential pharmacological effects owned to the peptides, enzymes, amino acids, and minerals contents. Bee venom possesses biological properties that are anticancer, anti-inflammatory, and antibacterial [71].

Global threats to human health and the environment are posed by methylmercury pollution. The etiology of Minamata sickness was determined to be caused by methylmercury, which is a particularly neurotoxic pollutant. The intake of seafood exposes people to methylmercury. Because methylmercury rapidly crosses the blood–brain barrier, it can have an impact on the neurological system [72]. *A. m. lamarckii* venom has an in vivo protective influence (apitherapy) against methyl mercury chloride-induced blood–brain barrier dysfunction and neurobehavioral toxicity [73].

As one of the most serious infectious illnesses affecting cattle, buffalo, swine, goats, and sheep, foot-and-mouth disease (FMD) is a major problem worldwide, as per its high transmissibility and related monetary and productive losses [74]. FMDV was treated with Egyptian bee venom. Bee venom injections administered daily for seven days to sick goats and guinea pigs resulted in a significant reduction in the viral load and a shorter treatment period [72].

The comparison between bee venom composition obtained from *A*. *m. lamarckii*, *A. m. carnica*, and the hybrid has been established, and demonstrated that the hybrid venom was more similar to *A*. *m. lamarckii* than *A. m. carnica*, reflecting the domination of *lamarkii* subspecies’ genetic characters over those of *carnica* [75].

Hyaluronidase, the main allergen in bee venom, contains four different *N*-linked carbohydrate sites, and consists of 373 amino acid residues. The enzyme acts as an anti-inflammatory by causing hyaluronic acid break down in various tissues, allowing it to enter the tissue. Treatment with hyaluronidase was reported to prevent tumor cells from reaching lymph nodes in T cell lymphoma, and it may have important anticancer effects to stifle tumor growth [76,77]. Recently, Abdel-Monsef and colleagues purified and characterized hyaluronidase enzyme for the first time from Egyptian bee venom homogeneously using DEAE-cellulose and Sephacryl S-300 columns [78,79].

Phospholipase A2 (PLA2), the second-largest component of bee venom after mellitin, is one of the most significant bee venom enzymes and might be regarded as one of its primary constituents. It is a lipolytic enzyme that hydrolyzes phospholipids at the sn-2-acyl linkage to release free fatty acids and lysophospholipids. PLA2 has a wide range of pharmacological properties, such as the ability to inhibit the growth of cancer cells and to have anti-inflammatory, hepatotoxic, and neuroprotective effects [80,81,82]. PLA2 enzyme was purified from *A. m. lamarckii* venom and displayed anti-coagulant and anti-platelet aggregation activities, making it a promising agent against clot formation [83].

Additionally, the bee venom of *A. m. lamarckii* can treat respiratory illnesses by improving the haematological and respiration parameters in the animal model [84].

## 8. Conclusions and Future Perspective 

*A. m. lamarckii* honeybee subspecies is thought to be the same subspecies as the one that could be found during the Pharaohs’ time, and it is still managed in mud-tube hives beside modern hives. The Egyptian *A. m. lamarckii* spreads in the Egyptian desert, delta region, Nile valley, and Sudan and resembles the lighter yellow-banded varieties with very dark drones. *A. m. lamarckii* is known for its excessive number of swarm cells, continuous breeding during the whole year, and low productivity, using very little propolis, and tends to migrate readily. *A. m. lamarckii* is characterized by higher mass-specific metabolism, presence of the half-queens, and high levels of defensive behaviors. The indigenous *A. m. lamarckii* subspecies have been hybridized with other introduced varieties, especially *A. m. carnica*; hence, pure subspecies have become very rare.

Due to the limited location and quantity of colonies in the small districts of Upper Egypt (Assiut), where it is kept in modern and mud tube hives, the populations of *A. m. lamarckii* have unfortunately suffered a significant decline. The owners of these Egyptian hives consider them a family heritage and refuse to accept governmental calls to rescue and enlarge them.

In order to maintain their colony stocks, beekeepers were compelled to purchase colonies of different subspecies, such as Italian and Carniolan honeybee. To date, the current status and the geographic variation of the *A. m. amarckii* populations in Egypt have not been studied in-depth and thus the data available is limited.

Futureward, efforts are highly recommended by the government, public, and researchers to preserve *A. m. lamarckii*, requiring more knowledge on the harmful effects threatening this bee subspecies as well as its conservation strategies, such as phylogenetic studies as well as the genetic network analysis. Moreover, beekeeping developmental projects can contribute to the conservation of honeybee and their environment. Qualitative and quantitative comparative studies could be conducted on *A. m. lamarckii* and the other Egyptian bee subspecies regarding their different products and how to be improved. One possibility that could be applied to improve the genetic conservation of local breeds in Egypt is the creation of isolated mating sites for the reproduction of *A. m. lamarckii* in the future breeding program. Although the Wadi Al-Assiut Protectorate uses the preserve for a range of flora and animals, including the native honeybees subspecies *A. m. lamarckii*, the protectorate should continue to focus on increasing bee populations with the help of the governmental scientific and financial support.

Further studies should focus on studying the *A. m. lamarckii* behavior that could positively affect the survival, efficiency, and decrease the infestation.

## Figures and Tables

**Figure 1 animals-12-02749-f001:**
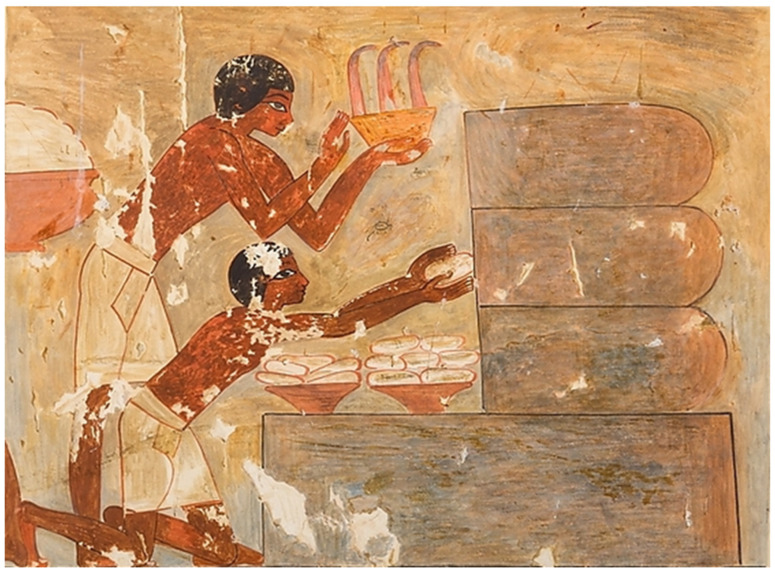
Honey combs gathering from large horizontal hives in Rekhmira tomb.

**Figure 2 animals-12-02749-f002:**
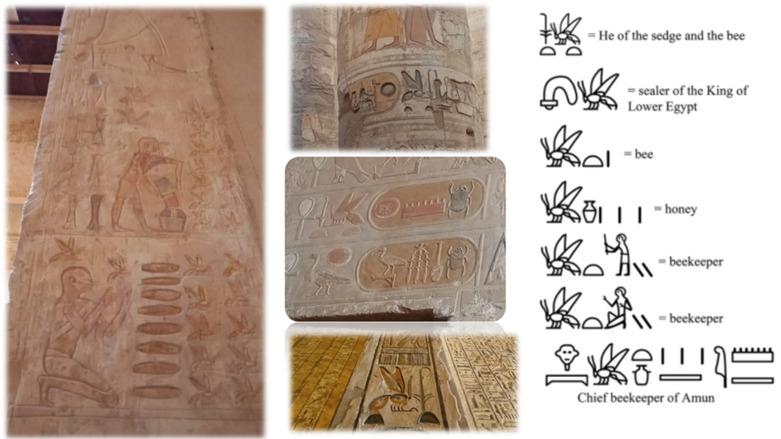
Beekeeping reliefs from the tomb of Pabasa and Karnak Temple (Photography by Aida Abd El-Wahed).

**Figure 3 animals-12-02749-f003:**
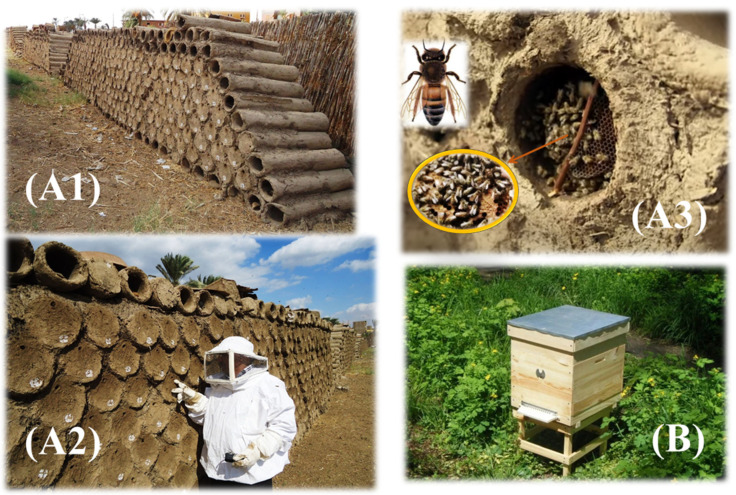
The Egyptian mud traditional hives (**A1**–**A3**) and modern ones (**B**). (Photo **A1**–**A3**: Dahy M. Mostafa and used with permission).

**Figure 4 animals-12-02749-f004:**
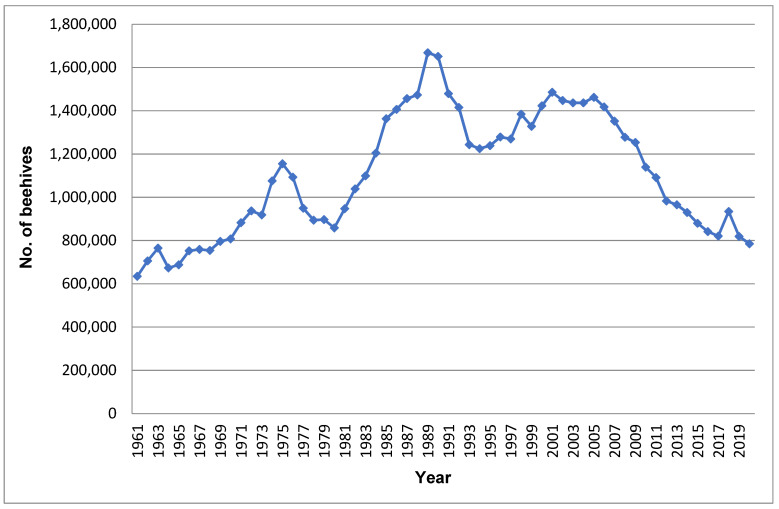
Beehives stocks in Egypt within the period (1961–2020) according to FAO (Data source: www.fao.org/faostat/en/#data/QCL, accessed on 29 July 2022).

**Table 1 animals-12-02749-t001:** Morphological characters of *Apis mellifera lamarckii* compared to *African Apis mellifera scutellata* and *Apis mellifera jemenitica*.

Morphology Characteristics	*Apis mellifera lamarckii*	*African Apis mellifera scutellata*	*Apis mellifera jemenitica*	Reference
Color	Lighter yellow-banded varieties with very dark drones	-	-	[7]
Width of worker cells	4.8 mm	-	-	[7]
Average head width	4.550 mm	-	-	[37]
Bodyweight	78 mg	-	87.65 mg	[11,40]
Body size	56.82 mm	55.04 mm	54.07 mm	[41]
Thorax mass	24 mg	-	-	[11]
Wing load	17 Nm^−2^	-	-	[11]
Wing size	45 mm^2^	-	-	[11]
Average length of forewing	5.575 mm	-	-	[37]
Forewing width	2.78–2.96 mm	2.71–3.03 mm	2.44, 3.20–3.03 and 2.77–3.23 mm	[35,36,42,43,44]
Leg size	7.39 mm	-	-	[36]
Forewing length	8.23 and 8.74 mm	8.48–9.01 mm	7.94, 7.53–9.01, 7.94, and 7.55–9.39 mm	[35,36,40,42,43,44,45,46]
Tongue length	5.75 mm	-	-	[35]
Hindwing length	6.11 mm	4.02–4.24 mm	5.85 mm	[35,40,41,42,45]
Hindwing width	1.76 mm	1.52–168 mm	1.67 mm	[35,42]
Femur length	2.24 mm	2.44–1.63 mm	2.44, 2.39–1.63 mm	[35,42,43]
Tibia length	2.82 mm	3.05–3.23 mm	2.55, 2.69–3.23 mm	[35,40,42,43]
Tibia width	-	-	1.12 mm	[40]
Basitarsus length	2.13 mm	1.81–2.02 mm	-	[35,42]
Basitarsus width	1.10 mm	1.05–1.15 mm	-	[35,42]
Cubital index	2.33 and 2.94 mm	1.77–2.86 & 2.33 mm	2.10–2.86 and 2.36 mm	[35,41,42,43,46]
Number of hooks	20.34	-	-	[35]
Lengths of the workers flagella	2.60–2.80 mm	-	2.29–2.80 mm	[38,43]
Lengths of the queens flagella	2.60 mm	-	-	[38]
Lengths of the half-queen flagella	2.48–2.60 mm	-	-	[38]
Length waxmirror	1.11 mm	1.16–1.32 mm	-	[36,42]
Waxmirror index	0.55 mm	-	-	[36]
Hair length	21.49 mm	20.88 mm	22.55 mm	[41]
Length of hind leg	738.09 mm	741.10 mm	717.71 mm	[41,43]
Wing angle J16	98.42	-	-	[41]
Proboscis length	5.65–570 mm	-	5.28, 5.30 and 4.84–5.74 mm	[40,43,45]
Metatarsus length	1.88–1.91 mm	-	2.03, 2.22 and 1.94–2.26 mm	[40,43,45]
Metatarsus width	1.05–1.08 mm	-	1.10, 1.01 and 0,97–1.16 mm	[40,43,45]
Wax mirror width	-	-	2.00 mm	[43]
Total length of antenna	3.92 mm	-	3.64 mm	[40,45]

(-): No reported.

## Data Availability

Not applicable.

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
