# Peer review of "A Spotlight on the Egyptian Honeybee (Apis mellifera lamarckii)"

_animals, 2022, doi:10.3390/ani12202749_

Round 1

Reviewer 1 Report

Dear authors,

Thank you very much for your effort to summarize through review till day published data on the honeybee species (Apis mellifera lamarckii).

My concerns are that you didn't quite well address the aim of your review. There is too little data and research used, only 47 references, of which not all are suitable for this paper.

You need to provide much more data, organize your paper better, and significantly expand all sections having in mind the importance of thematics.

Your review requires significant modifications, with fewer pictures and more tables, figures, and exact data.

I encourage you to revise your paper from beginning to end, modify it and resubmit it again.

All the best and stay safe

Author Response

Response letter enclosed 

Reviewer 2 Report

The Egyptian honeybee is a very important topic. Beekeeping is an important economic factor in Egypt, and has been for a very long time. Since a century, there is an ongoing tension field between the native subspecies (which is well-adapted to the environment), imported stock (more productive but vulnerable to disease) and hybrids (often problematic), as well as between traditional beekeeping and modern methods, and finally between utilization and conservation. Egyptian honeybee has unique characteristics and there is a growing interest in the scientific community. Thus, it is worthwhile to summarize the findings in a review.

However, parts of the review are so poorly written that it does not offer much additional value (also in comparison to the Introduction sections of several of the more recent publications cited). It is a pity that the work done in reviewing the publications is invalidated by an insufficient presentation.

Additionally, many of the facts are just listed. What I would expect from a "review" publication is the added value of connecting between the facts from the publications to a whole picture.

General remarks

1. Please name the species correctly! In many instances, including the title, it is more or less wrong. There are many valid options, from brief to detailed, such as:
Egyptian honeybee
Egyptian honey bee
Egyptian honeybee (Apis mellifera lamarckii)
Egyptian honeybee (Apis mellifera lamarckii Cockerell, 1906)
Apis mellifera lamarckii
A. m. lamarckii
Apis mellifera subsp. lamarckii Cockerell, 1906
Lamarck's honeybee
Lamarckii
Please write "Apis", "mellifera" and "lamarckii" in (sub)species names in italics, the rest in regular font.

2. There are too many orthographic, grammatical and typographic errors to mention them individually, some of them hamper understandability. Please rework thoroughly!

3. There is a very recent publication "Nafea et al. Ind J.o.Ent. 2022 Improving Egyptian Honeybee ..." worth including.

Individual criticism

Line 41. "Varro disorder", either Varrosis or Varroa disease
Line 41 "species" ( rather "subspecies" !) being just different is not worth mentioning - it is included in the notion of subspecies
Line 41 "found in Europe and is native to southern Egypt", it doesn't make sense this way. You probably want to draw attention to the fact, the A.m.jemenitica found in South Egypt is quite different albeit geographically neighbouring. European subspecies are different, this is trivial. The point is, European subspecies imported to Egypt differ much.

Line 53 "absconding" is the more bee specific term instead of "migrating"

Line 59 The Introduction goes straight into the history and far too small details (like the old taxonomic name). The purpose of the introduction is to highlight the importance of the subject (see my introductory sentences) and give only some cornerstones of information. There are repeated sentences from other sections (e.g. Conclusion). Instead, in the Introduction the sections should briefly be introduced.

Line 62 The relevance of Dakhla as well as the Asyut Governorate (not Government!) is not clear from the sentences. It is, however, very important and worth an explanation. I would suggest to open another section about critical events of the last century and the actual conservation status.

Line 84 The history section is interesting and well-written but it is out of balance in the context of the whole manuscript.

Line 128 This is a big jump from the previous text. What is missing is some details of the importation of European subspecies along the modern hives, the displacement of lamarckii, hybridization etc. which leads to the actual topic of the review.

Line 145 There are many absolute numbers given which are uninteresting. Instead, the comparison to other subspecies, relative numbers, are interesting. I suggest to collect the data in a table. Citation can go in a specific column. Another column can hold the average number throughout all subspecies (or whatever is given in the original publications).

Line 178 Near duplicate of the sentence starting 174

Line 188 "tolerant". As infestation levels are compared, "resistance" is the more appropriate term

Line 204 Mitochondria of different species (instead of subspecies) is a completely different topic. Mitochondria of subspecies show migration, human influence etc. in a completely different time scale. Just saying "far" is trivial and obsolete.

Line 206 [35] shows jemenitica in the same branch, so this statement that it is just similar to syriaca is a bit misleading as it suggests that the publications contradicts those who found close relationships to both syriaca and jemenitica.

Line 209 The question to be answered in this review is: do the studies more or less agree? Or are there contradictions? How does the genetic dis/similarity relate to geographical distances, different environmental conditions, impact of beekeeping?

Line 211 Rather than the name of the enzyme, the reader would like to know the implications from this test. Like, could it contribute to conservation efforts?

Line 214 Here information is just listed. But what does it mean, what are the consequences?

Line 227 Venom does not belong to the section "Genetic analysis". Perhaps, do a separate section on all venom studies. What are the conclusions?

Line 236 Detailed description of the findings. But what are the consequences? Are the Lamarckii more resistant to Varroa? And does this translate to better colony survival? And are, thus, the native honeybees an interesting alternative to imported European species despite their low yield and aggressiveness for beekeepers in Egypt?

Line 261 What is the proof, evidence and deduction chain? This would be a very interesting addition in the history section (which is completely subspecies-independent).

Line 272 "conservation strategies" - like what? This is basically what is missing in the manuscript: Where can A.m.lamarckii (apart from Asyut) be found? Are they endangered? By what? What is the beekeepers' position on this? I suggest a dedicated section "Current status and conservation".

Author Response

Response letter enclosed 

Reviewer 3 Report

The review article proposed by El-Seedi and colleagues aimed to be a specific view on the Apis mellifera lamarckii. 

Although the historical part is well organized, the remaining part is confusing and not detailed. 
Paragraphs 4, 5 and 6 are not exhaustive. They seem only a brief introduction of the issue discussed. Did the aim of this review be to characterize the subspecies lamarckii or only add very little information about the subspecies? 
In my opinion, a review article must be a complete art status about a specific argument.
the star point of this review is good, but some work is needed to improve and complete the article.

Author Response

Response letter enclosed 

Round 2

Reviewer 1 Report

Dear authors,

Thank you very much for taking the advice and suggestions, and making all corrections that have significantly improved the quality of your review. 

At this point, I'm satisfied with your review in its present form.

I suggest acceptance of your review after minor corrections.

References from the conclusions section have to be excluded or just relocated to another part of the text where appropriate. The conclusion part cant contain any references or citations.

After minor corrections, I suggest acceptance of this review paper for publication in the journal Animals.

All the best and stay safe

Author Response

Dear authors,

Thank you very much for taking the advice and suggestions, and making all corrections that have significantly improved the quality of your review. 

At this point, I'm satisfied with your review in its present form.

I suggest acceptance of your review after minor corrections.

References from the conclusions section have to be excluded or just relocated to another part of the text where appropriate. The conclusion part cant contain any references or citations.

After minor corrections, I suggest acceptance of this review paper for publication in the journal Animals.

All the best and stay safe

Response: Adjusted

Reviewer 3 Report

Authors reply to all my comments.
I have a few other comments:
-line 151. There are some references refering to table 1. However, in the table are reported different references. please revised.
- In the text are present some typos. Please re-check them to fix the sentence

Author Response

I have a few other comments:
-line 151. There are some references refering to table 1. However, in the table are reported different references. please revised.

Response: Adjusted
- In the text are present some typos. Please re-check them to fix the sentence

Response: Revised